# Spatiotemporal Regulation of Signaling: Focus on T Cell Activation and the Immunological Synapse

**DOI:** 10.3390/ijms21093283

**Published:** 2020-05-06

**Authors:** Esther Garcia, Shehab Ismail

**Affiliations:** CR-UK Beatson Institute, Garscube Estate, Switchback Road, Glasgow G61 1BD, UK

**Keywords:** signal transduction, T cells, immunological synapse, membrane domains, diffusion barriers

## Abstract

In a signaling network, not only the functions of molecules are important but when (temporal) and where (spatial) those functions are exerted and orchestrated is what defines the signaling output. To temporally and spatially modulate signaling events, cells generate specialized functional domains with variable lifetime and size that concentrate signaling molecules, enhancing their transduction potential. The plasma membrane is a key in this regulation, as it constitutes a primary signaling hub that integrates signals within and across the membrane. Here, we examine some of the mechanisms that cells exhibit to spatiotemporally regulate signal transduction, focusing on the early events of T cell activation from triggering of T cell receptor to formation and maturation of the immunological synapse.

## 1. Introduction

Complex organisms have evolved intricate molecular networks in which a single molecule can potentially participate in multiple pathways, performing signaling and/or structural roles. In such cellular networks, the functionality of molecules is defined by their structure, their potential interactors, and by their cellular context, that is, by their localization and time of action, which maximizes the potential capability of each molecule. Segregating signals spatiotemporally to control “when” and “how much” a stimulus is communicated is key for specificity and modulation, essential features of cellular signal transduction. Extracellular signals are transduced into the cell through the plasma membrane. On one hand, the plasma membrane acts as primary cell barrier, conferring self-containment and physical support to the cell. On the other hand, it constitutes a three-dimensional sensory and signaling platform that integrates signals in the lateral and axial planes [1], aiding spatiotemporal regulation.

T cells perfectly exemplify the spatiotemporal regulation of signaling. Despite the diversity in the molecular interactions between T cells and antigen presenting cells (APC) [2,3], some events are common to all of them. Several reviews have thoroughly discussed the steps that lead to T cell activation from various perspectives [4,5,6,7]. In brief, T cell activation takes place as follows: A migrating T cell screens its surrounding environment through active membrane protrusions, and, upon initial contact with an APC and successful recognition of the antigen presented, the area of contact between both cells increases rapidly (maximizing the area engaged in signaling). This initial contact evolves to a stable interaction that can last from minutes to hours, called the immunological synapse (IS) [8,9], after which the cells detach from each other (Figure 1a). At a molecular level, T cell activation involves the recognition of a cognate peptide presented by the major histocompatibility complex (pMHC) by the T cell receptor (TCR). TCR engagement leads to phosphorylation of the intracellular domains of the TCR-CD3 complex by the lymphocyte-specific protein tyrosine kinase LCK [10]. TCR-CD3 activation is tightly regulated by the activity and localization of LCK and the phosphatase CD45 [11,12]. TCR-CD3 activation leads to recruitment of the zeta-chain-associated protein kinase 70 (ZAP-70) and its subsequent phosphorylation by LCK [13]. ZAP-70 mediates recruitment of the linker for activation of T cells (LAT) that acts as docking site for the SH2 domain containing leukocyte protein 76 (SLP-76), phosphoinositide 3-kinase (PI3K), and phospholipase C-γ (PLC-γ) [14,15,16]. PLC-γ releases diacylglycerol (DAG) that activates the protein kinase C-θ (PKC-θ), which triggers later transcription of several factors such as the cytokine interleukin-2 (IL-2) and the receptor CD69 [17,18]. PI3K, in turn, produces the phosphatidylinositol PI(3,4,5)P_3_, leading to recruitment of several proteins involved in rearrangement of the actin cytoskeleton, essential for cell polarization and formation of the IS.

This sequence of events occurs at different timescales. The type of stimuli and sensor receiving and transducing the signal (mechanical or chemical) [20,21], the threshold and duration of triggering events [22], and the amplitude and amplification of the downstream signaling cascade [23,24] will determine the timescale of the signaling events. Thus, early T cell activation leading to IS formation occurs in three overlapping stages that progress at different time rates (Figure 1a) [19,25]. Firstly, the initial T cell-APC interaction requires an extremely fast rearrangement of the cell membrane and cytoskeleton that will evolve to a more stable interphase. During this stage, recognition of TCR and pMHC is essential, but mechanical cues also play an important role in the fast dynamics that occur in this phase. In a second stage, the area of contact between cells is maximized by changes in cell shape and an active relocation of molecules takes place within the interface, as well as new recruitment from other areas of the cell to the area of contact. During this phase, signal transduction mostly relies on amplification of signaling cascades and constant trafficking of signaling molecules to the interface and to recycling compartments. Finally, a third phase involves maturation and stabilization of the T cell-APC interaction, leading to formation of the IS. During this last phase, that lasts longer than the previous stages (Figure 1a), T cells distribute signaling events into domains, relying on several strategies of spatial regulation of signals. This confers structural stability while maintaining a sustained signaling.

Here, we aim to examine the mechanisms by which T cells modulate and segregate, signaling spatiotemporally, during early T cell activation and formation of the IS.

## 2. Temporal Regulation of Signaling during Early T Cell Activation

The very first stage of T cell activation is highly influenced by mechanical signals. Signaling events mediated by mechanical cues are rather fast, being able to communicate information in the order of nano- to milliseconds [20,21,26]. Mechanical signals diffuse rapidly, even at large distances, and can be transmitted directly to their target and last just as long as the stimulus is maintained, which allows a rapid on/off switch [20,27]. Mechanical forces spread across the plasma membrane via a range of systems including integrins and adhesion proteins in conjunction with the actin cytoskeleton, mechanosensitive receptors, and mechanosensitive ion channels. Transmission of these forces greatly depends on the biophysical properties of the plasma membrane including pH, electrostatic charges, membrane thickness, membrane fluidity, and membrane tension as well as on the properties of the stimuli [28,29,30]. For instance, mechanosensitive ion channels participate in regulating cell volume by sensing and responding to changes in tension and stretching of the plasma membrane, being directly linked to membrane elasticity and external stiffness (from either an interacting cell or from the extracellular matrix) [31].

The role of the actin cytoskeleton in mechanotransduction is well known, and the importance of actin during T cell activation has been thoroughly reviewed [32,33,34]. Yet, the very early steps prior to activation occur fast (from nanoseconds to few milliseconds), and many questions remain unanswered. To scan their environment, T cells develop filopodia or microvilli where they concentrate a plethora of sensing molecules: Receptors such as the TCR, adhesion proteins like the lymphocyte function-associated antigen 1 (LFA-1), ion channels, and other signaling proteins [35,36,37,38,39] (Figure 1b). These mechanosensitive protrusions represent the first contact with the APC. Filopodia are highly motile and present an acute membrane curvature at their distal tip. This together with their protrusive force and the mechanical pressure derived from the contact of the T cell and APC membranes may better expose the TCR and pMHC facilitating their encounter despite the thick cellular glycocalyx (50–500 nm, compared to the 4-nm-thick plasma membrane) of both T cell and APC [40,41] (Figure 1b). In addition to promoting TCR-peptide/MHC encounter [40,42], integrins like LFA-1 have an essential role in signal transduction. Upon TCR stimulation, LFA-1 undergoes a conformational change that increases its avidity to its ligand, the intercellular adhesion molecule 1 (ICAM-1). The conformation of LFA-1 changes to an even higher affinity conformation after interacting with ICAM1 [32]. Mechanical forces have a significant role in the molecular behavior of LFA-1, increasing the strength of their interaction with ICAM-1 as well as its bond lifetime [43]. Thus, in the absence of force, interactions of the same affinity require longer stimulation to obtain a similar cellular response. Interestingly, recurring exposure to force can prime adhesions, leading to a prolonged lifetime bond even after the force ceases, suggesting these molecules have some kind of “memory” mechanism [44].

Other mechanotransducers in T cells are mechanosensitive ion channels. These channels transduce changes in membrane tension and membrane stretching by altering their conformation and subsequently adjusting the transit of ions (Figure 1b). Thus, downregulation of the short transient potential receptor channel 3 (TPRC3) is linked to reduced proliferation of primary CD4+ T cells and reduced Ca^2+^ signaling in resting cells [45]. Another Ca^2+^ channel, the transient receptor potential cation channel subfamily V member 2 (TRPV2), showed lower response to membrane stretching when silenced in Jurkat (human lymphoma CD4+ cells) [46]. These studies have provided new targets to design drugs for pharmacological treatment of T cell disorders, and show the importance of mechanical cues and cellular volume sensors in T cell physiology.

The TCR is the main player in the initial steps that lead to T cell activation. The TCR has been traditionally studied from the perspective of ligand-receptor triggering, but in the last decade several groups have investigated its mechanosensitive potential. Integration of mechanical forces is only one aspect of TCR triggering, so we encourage the reader to consider two review articles that perfectly summarize the activation models proposed to date, and aim to combine the described triggering mechanisms into a unified model [47,48]. The TCR, but no other co-stimulatory receptors at the T cell membrane, can be activated by mechanical forces (originated from e.g., cytoskeletal forces at the dynamic T cell/APC [49]) and trigger T cell activation [50,51,52,53,54]. This adds a layer of regulation to TCR activation that has been overlooked until recently and provides further information regarding antigen sensitivity. Cellular binding forces originated from high affinity antigen-mediated TCR activation induce and sustain Ca^2+^ signaling and IL-2 secretion, increasing the sensitivity of T cell recognition by two orders of magnitude compared to low affinity antigens [55]. As observed for LFA-1, mechanical forces prolong the lifetime of the TCR-MHC interaction in the presence of agonist peptides, whereas in the presence of antagonist peptides the lifetime is actually shortened, suggesting that force not only lowers the threshold of activation but it also improves antigen discrimination [55,56,57].

T cells can trigger a cellular response that involves Ca^2+^ influx, phosphorylation of proteins, and production of cytokines by just one of the aforementioned mechanisms (TCR engagement, activation of integrins or ion channels, and even by contact with poly-L lysine [58]) or by just activating one single TCR-MHC complex [59,60]. Though redundancy is common in complex signaling networks and provides robustness to the system, all those signaling mechanisms in T cells might be necessary to deliver a highly specific and controlled response. Moreover, T cells do not show a true resting state. Their role is to scan for peptides and distinguish between self and foreign to avoid autoimmune responses, for which they go through a number of weak TCR-pMHC interactions (known as TCR tickling [61]). Thus, T cells rely on several mechanisms that contribute as co-stimulatory pathways to antigen discrimination but still allow fast response to foreign infections.

After initial triggering, cells modulate signaling by controlling the strength and duration of the transduction events. Signal strength seems to correlate in some T cell types (CD4+) with the duration of the interaction of T cell and APC [62], which, in turn, can determine cell fate [63]. The kinetic profiles of signaling molecules depend greatly on the activation threshold and immediate and future availability of ligand, receptor, and effector molecules. Thus, mechanisms such as TCR-pMHC affinity, receptor clustering, co-stimulatory factors, or recycling of signaling molecules (endocytosis/exocytosis events) can lead to different kinetic profiles or signaling waves during cell activation, leading to diverse stimulatory outcomes.

The activation threshold varies for each cell, stage of the immune response (naïve vs. activated T cells), and affinity and avidity for the antigen presented. T cells have developed a system of positive and negative co-stimulatory receptors to modulate their responsiveness to antigen, which is the case of CD28 (positive) and CTLA-4 (negative) that can regulate the threshold needed for activation [64,65] (Figure 1c). LCK was also reported to reduce the activation threshold upon TCR triggering in naïve T cells, leading to up to a 100-fold shift in response to increasing doses of anti-CD3 when compared to LCK knock-out cells reconstituted with FYN [66].

Brockmeyer et al. described how TCR activation leads to at least three distinctive phosphorylation kinetic profiles or signaling waves [67]. In a first wave, that peaks at ~30 s, the authors detected phosphorylation of TCR and the nearest proteins: ZAP-70, LAT, SLP-76, PLC-γ1, interleukin-2-inducible T-cell kinase (ITK), VAV-1, and the tyrosine-protein phosphatase nonreceptor type 6 (PTPN6) (Figure 1c). Interestingly, LCK showed no significant increase in phosphorylation of its tyrosine 394 compared to its basal state. This is unsurprising considering the studies showing that LCK is constitutively active prior to TCR triggering (although the levels of pre-activated LCK vary from 2% to 50% [68,69,70]). A second activation peak at ~2 min included the engulfment and cell motility protein 1 (ELMO-1), the dedicator of cytokinesis (DOCK), GIT-1 or COOL-2 that recruit proteins like Rac2, the P21-activated kinase 1 (PAK1) and PLC-γ1 and, therefore, play a role in regulating actin cytoskeleton dynamics [67] (Figure 1c). A third peak, between 5 and 10 min, comprised components of clathrin-coated vesicles and endocytic compartments [67] (Figure 1c). Whether these kinetic profiles result from an initial TCR triggering or sustained engagement is unclear, yet TCR dynamics and localization throughout the activation process suggest that TCR signaling needs to be sustained over time, especially in the case of naïve cytotoxic T cells that require continued signaling of several hours to be activated [71]. To sustain or downregulate signaling, T cells resort to endocytosis and recycling of molecules back to the plasma membrane. Engaged TCR is quickly endocytosed, consequently needing fast recycle of receptors back to the plasma membrane. TCR-pMHC complexes are internalized in a clathrin-independent manner (Figure 1c) and recycled back via Rab4, Rab5, and Rab11 endosomes, for which the actin and tubulin cytoskeletons are essential [72,73,74,75].

LCK is a key player during T cell activation. Despite having been studied in the context of early activation, how it localizes and phosphorylates remain unclear. During the initial stages of activation, a first signaling wave by active LCK phosphorylates CD3 immunoreceptor tyrosine-based activation motifs (ITAMs) upon TCR triggering (Figure 1c). After TCR triggering and IS formation, CD3, CD4, CD28, CD45, and LCK translocate to the IS, where LCK then phosphorylates CD3 ITAMs anew at the periphery of the newly formed synapse. Since LCK continuously localizes to the IS, a sustained transport of LCK to the plasma membrane is necessary. LCK is recycled back to the plasma membrane in a process that requires Rab11 and the Rab11 family-interacting protein 3 (FIP3) system (Figure 1c), the late endosomal transporter CD222, and the uncoordinated 119 protein (UNC119, a guanosine diphosphate dissociation inhibitor (GDI)-like solubilizing factor) (Figure 1c). Moreover, the lipid raft-associated myelin and lymphocytes protein MAL is responsible of targeted traffic of LCK to lipid rafts [76,77,78]. UNC119 focuses LCK at the IS and it is required for IS formation [78,79]. UNC119 extracts LCK from membranes by sequestering its myristoyl group and releasing it at a target membrane through a mechanism regulated by the small GTPase adenosine diphosphate-ribosylation factor-like protein 3 (ARL3) and its guanine nucleotide exchange factor, ARL13B [79,80] (Figure 1c). We hypothesized that LCK could be extracted from endomembranes (such as Rab11 endosomes) and/or from nonsynaptic areas of the plasma membrane and placed at the IS. UNC119-mediated traffic of LCK would potentially reduce fusion of vesicles with the plasma membrane at the IS, contributing to membrane homeostasis and preservation of specialized membrane domains. This mechanism is analogous to that of the phosphodiesterase 6δ (PDE6δ) and its cargo, the inositol polyphosphate 5-phosphatase E (INPP5E) at the primary cilia [81]. The transport of proteins into the cilium is very restricted and highly dependent on vesicles being delivered to the base of the cilium. It is, therefore, likely that IS and primary cilia share recycling routes and parallel solubilizing factors.

Similar to TCR and LCK, LAT exists as two different pools in T cells: One at the plasma membrane and one intracellular. LAT is internalized in a clathrin-dependent manner (Figure 1c), going through the Golgi Apparatus before being recycled back to the IS [82]. LAT localizes to Rab27 and Rab37 endosomes delivered to the IS via an IFT20, v-SNAP Receptor (v-SNARE), and Vesicle Associated Membrane Protein 7 (VAMP-7) dependent process [83,84,85]. Highly time-resolved microscopy experiments by Balagopalan et al. have elegantly provided new hints on how LAT is recruited to the IS during T cell activation. The authors found that LAT microclusters firstly form by recruiting LAT via lateral diffusion, without contribution from the intracellular pool. VAMP-7 vesicles containing LAT were only recruited to the IS later on, after microcluster formation. Interestingly, LAT vesicles moved between clusters (and interacted with them) via microtubule-associated transport [86].

These few examples stress the relevance of how key molecules are delivered to the plasma membrane and activated over time during T cell activation. Although endocytosis of exhausted (not able to transduce signals anymore) and non-exhausted signaling molecules could initially contribute to downregulation of signaling, growing evidence suggests that active signaling occurs in endosomes of T cells [87,88]. This increases the complexity of the signaling sequence taking place in the T cell, and adds a new signaling compartment, expanding the role of the plasma membrane as a signaling hub. To fully comprehend the early stages of T cell activation, we need more studies like Balagopalan’s, focused on earlier time points and high temporal resolution, expanding our view from the plasma membrane to other membrane compartments as tools for spatial regulation.

## 3. Spatial Regulation of Signaling during Early T Cell Activation

Compartmentalization aids sorting of molecules to perform different tasks. Cellular compartments include organelles or, at a much smaller scale, specialized functional domains. These domains can be found in the cytoplasm as, for instance, self-contained lipid droplets, but they have been mostly depicted in the plasma membrane. Membrane domains can range in size from microns (such as filopodia, the neurological and immunological synapses or the primary cilia) to nanometers, as in the case of highly ordered lipid domains or “lipid rafts”. Most of these specialized domains share a relatively high concentration of receptors, adhesion proteins, and other signaling molecules, as well as a particular lipid composition; but their lifetime might differ, leading to more transient or stable domains. Membrane domains require a certain level of stability to keep their functionality over time. Thus, cells resort to several mechanisms to preserve the structure and composition of membrane domains: From directing vesicle trafficking to the domain, to heterogeneously distributing specific lipids and proteins/enzymes, or reducing the mobility of molecules.

The IS is a clear example of spatial regulation. The structure of the IS varies depending on the type of T cells and APCs involved (primary, cooperative, or cytotoxic T cells), their stage of maturation, and the outcome of their interaction [2,3,89]. For instance, CD3ζ localization pattern differs between thymocytes and mature T cells [90]. The most commonly studied IS is that of CD4+ T cells interacting with lymphoma B cells or cytotoxic T cells, that mature into a bull’s eye-like structure formed by three concentrical rings termed supramacromolecular activation centers (SMACs): The cSMAC at the center; the pSMAC, a peripheral ring that surrounds the cSMAC; and the dSMAC, at the edge/distal area of the synapse [8] (Figure 2a). The dSMAC develops from an actin ring [9,91] (Figure 2a) that will first expand and later retract in a process dependent on nonmuscular myosin IIA [92,93].

Throughout the maturation of the IS, signaling molecules are sorted into different SMACs. After initial triggering, TCRs at the plasma membrane diffuse laterally across the membrane and aggregate, forming multiple microclusters [19]. Initially, TCR microclusters are commonly surrounded by an LFA-1 adhesion ring, leading to more stable synapses, extremely relevant in the presence of low affinity ligands [94]. TCR clusters move from the periphery to the center of the newly formed IS (future cSMAC) (Figure 2a). The translocation of TCR to the cSMAC was initially interpreted as a way of accumulating signaling in that area, but other studies suggest a more complex mechanism that sorts molecules in different areas of the IS [95]. LCK concentrates at the dSMAC instead of the cSMAC [96], which challenges the idea of sustained TCR signaling at the cSMAC. While CD45 is early excluded from stable TCR microclusters and segregates to the pSMAC, sustained TCR signaling seems dependent on CD45 translocating to the cSMAC to reset TCR signaling (where 93% of cSMAC clusters contained CD45) [95,97]. Moreover, though the co-stimulatory receptor CD28 goes to the cSMAC, it localizes to its periphery instead of areas of TCR concentration, suggesting that cSMAC TCR consists of exhausted receptors heading to recycling. Yet, in the case of strong antigenic stimuli, cSMAC TCR has been linked to sustained active signaling [98]. Though the distribution of TCR is neither universal nor directly connected to its activity, it plays an important role in regulating signaling.

### 3.1. Generation of Specialized Membrane Domains at the Immunological Synapse

Targeted vesicle trafficking to specific areas of the plasma membrane generates membrane domains with a unique protein composition. This is a common strategy in cells with very active and sustained signaling, mediated by ligand–receptor interactions that rely on a high recycling rate. T cells polarize their exocytic machinery towards the IS by reorganizing the actin and tubulin cytoskeletons. The centrosome, that is associated to the Golgi Apparatus and centralizes the vesicle traffic of the cell, relocates underneath the IS [99,100], accelerating the recycling pathways. Centrosome relocation is mediated by the lipid DAG, in response to TCR stimulation [101].

Molecules recruited to the IS are commonly targeted to specific membrane domains. It is unclear whether TCR nanoclusters localize to highly ordered lipid domains prior to activation [102,103] but most studies agree that TCR concentrates at these domains upon triggering [104,105], contributing to lowering its activation threshold [106]. Other relevant signaling proteins such as LCK, FYN, or LAT are also associated to highly ordered lipid domains [107], and their lipid modifications are essential for their localization to the plasma membrane [79,108] (Figure 2b). Surprisingly, the pool of active LCK seems excluded from highly ordered lipid domains, which stresses the importance of spatial regulation in modulating signaling [109,110]. It is likely that highly ordered lipid domains act as docking platforms that localize and integrate signal transduction. Additional observations support this hypothesis: The lipid raft-associated protein MAL not only acts as a lipid-packing molecule, likely due to its multiple transmembrane domains, but also localizes with LCK in endosomes and it is essential for its translocation back to the plasma membrane [76,111]. As several other proteins, MAL plays similar functions at the IS and the primary cilia. MAL contributes to maintaining the condensed membranes at the base of the cilium and lack of MAL results in mislocalization of Rab8, the intraflagellar transport proteins IFT20 and IFT88, and smoothened from the cilia to the ciliary base [112]. Highly ordered lipid domains play an important role in primary cilia as well, possibly due to their contribution to the specific placement of lipid-modified proteins at the cilium.

Specialized membrane domains can be originated by allocating enzymes with antagonistic activities in strategic sites, which leads to compartments composed by gradients of different enzymatic products (Figure 2a,b). The ratio of these gradients will depend not only on the concentration and distribution of these enzymes but also on the molecular diffusion of the enzymatic products, thus it is tightly linked to the physical properties of the plasma membrane. An excellent example of this, is the heterogenous distribution of PI3K-PLC-γ and the phosphatidylinositol-4-phosphate 5-kinase (PIP5K) across the plasma membrane at the IS [113,114]. PI3K and PLC-γ metabolize the PI(4,5)P_2_ produced by PIP5K. Redistribution of these kinases during T cell activation leads to drastic changes in localization of PI(4,5)P_2_ and, subsequently, in the activity of PI(4,5)P_2_ effectors. After TCR engagement, PLC-γ moves to activated TCR areas, reducing the levels of PI(4,5)P_2,_ which is, in turn, replaced by DAG [113,115]. Since PI(4,5)P_2_ directly interacts with several actin-related proteins (such as ezrin, radixin, and moesin (ERM) proteins) or the nucleating-promotor factor, Wiskott–Aldrich Syndrome Protein (WASP)), its redistribution has a striking effect on actin cytoskeleton dynamics. Thus, loss of PI(4,5)P_2_ leads to clearance of filamentous actin from the center of the IS [114] (Figure 2a). In addition to its clearance from the central synapse, PI(3,4,5)P_3_ also drives the formation of the F-actin ring (Figure 2a). In a process mediated by Ras and PI3K, annular distribution of PI(3,4,5)P_3_ precedes and leads the formation of the contractile actin ring in cytotoxic T cells [116]. A similar system discriminates membrane domains at the primary cilia. At the cilium, INPP5E converts PI(4,5)P_2_ into PI(4)P, whereas INPP5F/OCRL1 metabolizes PI(3,4,5)P_3_ down to PI(3,4)P_2_ [117,118]. These two proteins localize to different areas of the cilium (a process that is tightly regulated by PDE6δ and ARL proteins [80]), leading to a ciliary membrane rich in PI(4)P and PI(3,4)P_2,_ while the plasma membrane surrounding the cilium is characterized by higher levels of PI(3,4,5)P_3_ and PI(4,5)P_2_ [119,120,121,122]. As in the IS, increase of PI(4,5)P_2_ mediates actin recruitment that could interfere with vesicle traffic to the cilia [113]. Phosphatidylinositol gradients not only generate domains at the plasma membrane but throughout all cell membranes [123].

### 3.2. Dynamic Behavior of Molecules at the Immunological Synapse: Barriers to Free Diffusion

In highly ordered lipid domains or lipid rafts, where content of cholesterol and sphingomyelins is higher than in the surrounding membrane and contains highly packed lipids, proteins diffuse slower than outside of the domain, increasing the potential interaction between proteins localized to the same domain (Figure 2b). Similarly, protein clusters and protein islands move at an inversely proportional speed to their size, and obstruct the movement of neighboring molecules [124]. The high concentration of molecules in these domains affects not only the molecular diffusion inside the domain, but also its translational movement (when limited to free lateral diffusion). The concentration of signaling molecules together with their limited mobility facilitates potential interactions, strengthening the likely role of membrane domains as signaling hubs. However, it is not uncommon that protein clusters move faster than their diffusive potential, due to active transport mediated by the actin cytoskeleton [125,126]. We can find examples of both diffusive and active transport of protein clusters across the synapse, but the contribution of each of these components to their overall mobility is still unclear. TCR clusters, for instance, rely on actin retrograde transport to translocate from the edge of the synapse to its center. However, experiments using latrunculin B showed that only 50% of the TCR pool was attached to actin, since a population of clustered TCR was still able to reach the center of the IS despite the treatment [127]. Similar results were obtained after latrunculin A treatment, where of 80% of pMHC and TCR signals was maintained at the cSMAC [95]. Other studies suggest that the size of TCR clusters is key to their translocation dynamics, as TCR clusters will interact with actin with variable strength depending on the number of molecules involved [125,128], which would explain the different values reported.

All the energy cells employ to generate specialized domains would be inefficient if molecules diffused freely after being targeted to a specific domain. Thus, diffusion barriers contribute to maintain a stable molecular composition over time. Proteins of the cortical actin cytoskeleton such as spectrins, ankyrin, and occasionally adhesion proteins, can interfere with the lateral diffusion of membrane components (reviewed in [129]) (Figure 2b). Similarly, interactions between transmembrane proteins and the actin cytoskeleton can act as picket fences that constrain movement of molecules to areas of confinement in between these picket fences [130,131] (Figure 2b). Yet, some mobility is allowed between these confinement areas (i.e., hopping events), but the mechanisms are still unclear [129,130,132,133,134]. We believe that controlled leakage across diffusion barriers at the IS adds a layer of regulation to T cell signaling. Based on our data on UNC119-mediated trafficking of LCK to the IS [79], we speculate that UNC119 could aid LCK lateral diffusion at the plasma membrane and overcome diffusion barriers by relocating LCK between membrane compartments (e.g., from non-IS to IS). A recent study showed that UNC119 forms a complex with Uncoordinated (UNC)-44 (ankyrin) and UNC-33 (or CRMP) in *C. elegans*’ primary neurons [135], linking UNC119 with the cortical actin, which would support a potential role for UNC119 in translocating molecules across membrane domains.

Although there is still little evidence that links the cSMAC, pSMAC, and dSMAC compartments and the contractile actin ring, we believe that the actin ring potentially acts as a picket fence to impair free diffusion between the SMACs. Translocation of ICAM1 relies on actin retrograde flow from the edge of the synapse to inner areas, but ICAM1 is restricted from the mature cSMAC, supporting the existence of a barrier that limits access of ICAM1 to the center of the synapse [127]. This barrier leaks some molecules like TCR that, as we discussed before, can localize to the cSMAC in an actin-dependent and -independent way [127]. Highly ordered lipid domains accumulate at peripheral regions of the synapse, often in an annular distribution [136]. There is some controversy regarding whether condensed lipid domains are found at the central synapse or not, which is possibly sustained by the different microscopy techniques used in these studies: No central condensed lipid domains were detected by total internal reflection fluorescence (TIRF) microscopy [136], whereas confocal microscopy data reported central localization of these domains [137,138]. The different axial depth of these techniques might explain the contrasting data, but all these results are consistent since confocal datasets might be detecting vesicle trafficking events near the plasma membrane that TIRF omits. This leaves open the debate of a diffusion barrier between the dSMAC/pSMAC and cSMAC, and whether the contractile actin ring and annular lipid domains are linked. Attachment between the actin ring and annular lipid domains would likely pose a diffusion barrier that would restrict molecules crossing in and out of the annular structure and would support the spatial regulation associated to the SMACs. There is need for more studies that tackle the intertwined dynamics of the lipids of the plasma membrane and the actin cytoskeleton.

## 4. Conclusion and Future Perspectives

We have examined the relevance of signaling dynamics, focusing on T cells since their fast and organized activation utilize a wide range of mechanisms for signal regulation. T cells have developed a multitude of tools to speed up signaling without compromising specificity; and have efficiently exploited signaling compartments and functional membrane domains to combine stability with fast recycling and interchange of molecules to keep signaling active over time.

A clear conclusion is that all the mechanisms of spatiotemporal regulation described here work together in tight regulation. Ciliary proteins, like UNC119 and its regulators, have been overlooked in the context of T cell activation until very recently. Although we still do not fully understand the role that these proteins might play in the temporal and spatial regulation of T cell signaling, we believe they might be adding a layer of regulation to LCK activity and localization aside from their recruitment by lateral diffusion and vesicular trafficking. To address this, future studies should focus on obtaining reliable quantitative information of the molecular dynamics of lipid-modified proteins like LCK and their solubilizing factors during the very early stages of T cell activation. It is worth investigating further the contribution of temporal and spatial regulation of signaling to the immunity of individuals. Are these pathways redundant and so interconnected that the cell bypasses any potential defect or deficiency? Most studies have focused on the importance of the targeting of signaling proteins at the synapse in T cell response, but little is known regarding the importance of a timely response. How do changes in the speed of the early T cell response affect immunity? There is need for further studies that focus on the dynamics of signaling in immunity.

## Figures and Tables

**Figure 1 ijms-21-03283-f001:**
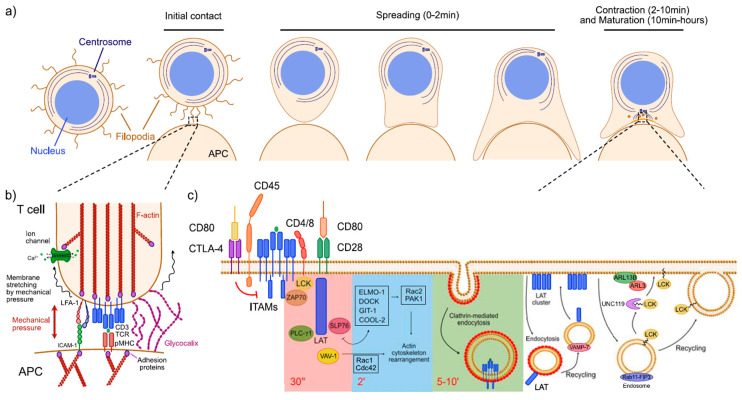
Temporal regulation of signaling during early T cell activation. (**a**) The schematic shows the stages of early T cell activation and immunological synapse formation (adapted from [19]). T cells scan their environment via filopodia sensing. After initial contact (that occurs within seconds), the T cell spreads rapidly, increasing the area of contact between the T cell and antigen-presenting cell (APC) (which lasts up to 2 min). After full T-cell spreading, the T cell retracts, in a process that involves actin cytoskeleton rearrangement aided by adhesion proteins and proteins that mediate contractility (2–10 min). Cell contraction is accompanied by an active retrograde transport of signaling molecules from the periphery of the interaction to the center that will ultimately lead to a mature interaction and formation of the immunological synapse. (**b**) Schematic of initial contact between the T cell and APC mediated by filopodia/microvilli. Filopodia act as primary cell sensors, concentrating integrins, receptors, ion channels, kinases, and phosphatases in other areas of the plasma membrane. Upon contact with an APC, an array of mechanical and chemical signals involving the T cell receptor-Major Histocompatibility Complex (TCR-MHC) and the Lymphocyte function-associated antigen 1-Intercellular Adhesion Molecule 1 (LFA-1-ICAM-1) interactions (among others), rapidly lead to drastic rearrangement of the membrane and the actin cytoskeleton. The thick cellular glycocalyx might be compressed and even displaced upon contact, allowing a better access to receptors and ligands from T cell and APC, increasing the chances of encounter. The TCR is sensitive to mechanical forces that alter its sensitivity and specificity to ligand, modifying the threshold of activation. As other integrins, LFA-1 is involved in transducing mechanical information into the cell through adhesion proteins, leading to changes in the actin cytoskeleton. Mechanical forces also determine the lifetime bond of LFA-1 and its ligand, leading to interactions of different strength and duration. Mechanosensitive ion channels or transient receptor potential proteins detect stretching and changes in tension in the near membrane altering their conformation and modifying the transit of ions that will initiate signaling cascades involved in actin cytoskeleton rearrangement and T cell activation. (**c**) The schematic shows the kinetics of TCR, LCK and LAT upon initial TCR triggering. Co-stimulatory receptors CTLA-4 and CD28 regulate the threshold of TCR activation, enhancing or downregulating activation. The kinetic profile of TCR-mediated signaling corresponds to three signaling waves: (1) the zeta-chain-associated protein kinase 70 (ZAP-70), the linker for activation of T cells (LAT), the SH2 domain containing leukocyte protein 76 (SLP-76), phospholipase C-γ (PLC-γ1) and VAV-1 are activated within 30 s of initial triggering; (2) at around 2 min a second group of proteins involved in recruiting actin cytoskeleton factors are activated; and (3) after 5–10 min machinery involved in clathrin-mediated endocytosis gets activated. LCK-mediated TCR-CD3 activation is negatively regulated by CD45 and positively modulated by CD4 and CD8. LCK membrane levels are regulated by Rab11-FIP3 endosomes and the solubilizing factor Uncoordinated 119 (UNC119) that mediates its recycling back to the plasma membrane. LAT is recruited to the plasma membrane in two waves: (1) LAT is recruited by lateral diffusion forming clusters and (2) the intracellular pool is recruited via Vesicle Associated Membrane Protein 7 (VAMP-7) vesicles to the plasma membrane.

**Figure 2 ijms-21-03283-f002:**
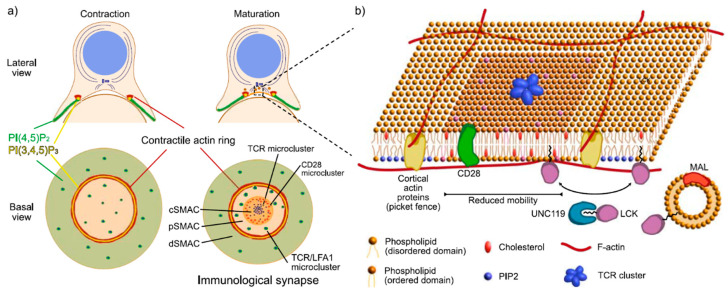
Spatial regulation of signaling at the immunological synapse. (**a**) The schematic shows maturation of the T cell-APC interaction to the IS. The lateral view shows the stage of polarization of the T cell: The centrosome is recruited underneath the cell–cell interphase and there is active transport of proteins and vesicles towards the APC, while the nucleus moves to the rear area of the cell. To mediate cell contraction and retrograde flow of molecules to the center of the interphase, a contractile actin ring develops which coincides with an annular distribution of PI(3,4,5)P_3_. PI(4,5)P_2_ and PI(3,4,5)P_3_ regulate actin clearance from the center of the synapse. After maturation and formation of the IS, signaling molecules are sorted into the supramolecular activation centers (SMACs): central (c), peripheral (p), and distal (d) SMACs. Clusters of TCR, CD28, and TCR combined with other signaling molecules like LFA-1 are sorted into different SMACs during the maturation of the synapse. (**b**) The schematic shows some of the strategies cells rely on to generate specialized membrane domains and how they influence mobility of molecules within and between domains. Highly ordered lipid domains contain a higher percentage of saturated fatty acids, sphingomyelin, and cholesterol, as well as highly packed phospholipids. Distribution of phosphatidylinositols such as PI(4,5)P_2_ and PI(3,4,5)P_3_ also generates membrane domains that lead to a distinctive interaction of the membrane and the actin cytoskeleton. Actin filaments together with cortical actin proteins such as ankyrin or spectrins as well as transmembrane proteins generate barriers to free diffusion of molecules that act as picket fences, creating confinement of molecules within the compartment. TCR clusters and other protein clusters frequently localize to highly ordered lipid domains, since assembly of lipid and protein clusters are closely connected. Several proteins are specifically localized to highly ordered lipid domains, usually through lipid modifications (myristoilation, palmytoilation, etc.); such is the case of LCK that is myristoilated and targeted to condensed domains by myelin and lymphocytes protein (MAL)-positive vesicles. LCK might translocate from vesicles to the plasma membrane and between membrane domains (overcoming potential diffusion barriers) by UNC119.

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
