# Peer review of "Spatiotemporal Regulation of Signaling: Focus on T Cell Activation and the Immunological Synapse"

_ijms, 2020, doi:10.3390/ijms21093283_

Round 1

Reviewer 1 Report

The manuscript presented by Garcia and Ismail attempts to review the spatiotemporal segregation of signaling during the formation of the immunological synapse after T cell activation, however the text needs an extended revision and major rewriting for further consideration.

Major:

As a general commentary, the article contains many concepts and ideas written in a disorganized manner, the body of the text has no clear story line, the players of the manuscript are not properly presented and there is repetition of ideas throughout the manuscript. In addition, there is no clear highlight of the novel contributions on the specific field as well as the scientific questions that should be further addressed.

The article contains unnecessary text that makes the reading very difficult to follow. For example, the initial sentence of the abstract is hard to understand and generates doubts about the direction of the manuscript. Then, the introductory paragraphs display very general ideas with very few references (if any) that make the reader wonder about the focus of the article.

The concepts and ideas of the article are disconnected. Many notions are assumed as previous knowledge in the initial paragraphs but then these concepts are later presented as new. Some examples among many; lines 52 to 66 present many key molecules with abbreviations with no proper presentation of these players. For example LCK is called in page 2, but it is presented/described in page 6. The concept of immunological synapse is presented in page 6, line 202, but it was already mentioned before, even with a full figure (Fig. 2). The same happens with the presentation of supramolecular activation clusters (SMACs), which are presented in page 7, line 256, when they were already mentioned in fig. 2. GDI is presented in page 9, but it was already mentioned in figure 2. The centrosome relocation is mentioned in page 7, line 282, when it was already mentioned as MTOC in the previous page. In page 9, line 371, the segregation of proteins such us TCR and LFA-1 are described as first, when SMACs have been already discussed. This makes really difficult to follow the story line of this manuscript.

Basically, this article needs a full revision re-focused on T cell activation and immunological synapse formation. In the view of this referee, the manuscript should start with the thorough description of the T cell immunological synapse formation during T cell activation, properly and orderly describing the molecular players in this scenario, and then highlighting the specific contributions on the ciliary machinery and temporal segregation of signaling.

Minor:

Some sentences are hard to follow in the context (for example, “At bird’s view”, supposedly “bird’s-eye view”, or the expression “speed of sound”) and need rewriting in a more scientific (less journalistic) context.

It is hard to understand why some words or sentences are underlined or bold.

In line 279, immunological synapse should be already written as the abbreviation IS, which was described in line 202. The manuscript should be carefully edited. 

Author Response

The manuscript presented by Garcia and Ismail attempts to review the spatiotemporal segregation of signaling during the formation of the immunological synapse after T cell activation, however the text needs an extended revision and major rewriting for further consideration.

Major:

As a general commentary, the article contains many concepts and ideas written in a disorganized manner, the body of the text has no clear story line, the players of the manuscript are not properly presented and there is repetition of ideas throughout the manuscript. In addition, there is no clear highlight of the novel contributions on the specific field as well as the scientific questions that should be further addressed.

Response: We appreciate the reviewer suggestions. To address these concerns, we have edited the manuscript as follows:

  • We have removed non-essential information from the original text to make it more simple and clearer and have restructured the contents of the manuscript in a more organised fashion. We have stablished more clear sections and subsections (sections 3.1 and 3.2) to organise the ideas more clearly.
  • To establish a clearer story line, we have expanded the text in some sections (like the introduction) in order to clarify the purpose of the review and the main ideas of the manuscript. We have removed some information of some sections (section 3) to facilitate understanding of the main ideas of the manuscript.
  • To better introduce the main signalling players, we have presented them in the introduction and expanded the description of the signalling events that lead to T cell activation. All molecules are now appropriately described and their acronyms defined from the very beginning.
  • By reorganising the text, we have removed repetition of ideas making the manuscript more engaging and compelling.
  • To highlight novel contributions, we have developed more clearly some of the ideas and questions discussed in the text and their impact in the field.

The article contains unnecessary text that makes the reading very difficult to follow. For example, the initial sentence of the abstract is hard to understand and generates doubts about the direction of the manuscript. Then, the introductory paragraphs display very general ideas with very few references (if any) that make the reader wonder about the focus of the article.

Response: we appreciate the constructive comment made by the reviewer. To address this, we have performed the following modifications:

  • We have removed unnecessary text throughout the manuscript to make it easier to follow. In particular, we have modified the abstract, the introduction, and have reduced considerably the text in section 3.
  • To better depict the focus of the article we have increased the number of references and tried to be clearer and more specific in the way we present the main ideas.

The concepts and ideas of the article are disconnected. Many notions are assumed as previous knowledge in the initial paragraphs but then these concepts are later presented as new. Some examples among many; lines 52 to 66 present many key molecules with abbreviations with no proper presentation of these players. For example LCK is called in page 2, but it is presented/described in page 6. The concept of immunological synapse is presented in page 6, line 202, but it was already mentioned before, even with a full figure (Fig. 2). The same happens with the presentation of supramolecular activation clusters (SMACs), which are presented in page 7, line 256, when they were already mentioned in fig. 2. GDI is presented in page 9, but it was already mentioned in figure 2. The centrosome relocation is mentioned in page 7, line 282, when it was already mentioned as MTOC in the previous page. In page 9, line 371, the segregation of proteins such us TCR and LFA-1 are described as first, when SMACs have been already discussed. This makes really difficult to follow the story line of this manuscript.

Response: We agree have made the following changes to present the ideas in a more connected way:

  • We have carefully reviewed that all players are properly presented now in the introduction and removed any repetition of ideas to avoid confusion. More specifically:
    • All abbreviations are now properly presented and followed up later on.
    • LCK is now properly presented in the introduction with the other players, and then discussed in sections 2 and 3.
    • Immunological synapse is now only presented once and followed up later on by the acronym IS.
    • SMACs are presented in section 3, and furtherly discussed in sections 3.1 and 3.2.
    • GDI is now firstly presented in the text and later in the new figure 2.
    • TCR and LFA-1 spatial segregation is addressed after presentation of SMACs (section 3) in sections 3.1, 3.2.

Basically, this article needs a full revision re-focused on T cell activation and immunological synapse formation. In the view of this referee, the manuscript should start with the thorough description of the T cell immunological synapse formation during T cell activation, properly and orderly describing the molecular players in this scenario, and then highlighting the specific contributions on the ciliary machinery and temporal segregation of signaling.

Response: we appreciate the time and considerations of the reviewer to improve the manuscript. We believe we have addressed the reviewer suggestions as follows:

  • We have re-focused T cell activation and immunological synapse formation by thoroughly and orderly describing in the introduction players involved in the signalling steps to T cell activation and IS formation and later systematically addressing the diverse strategies to temporal and spatial regulation of those players during early activation.
  • We have also highlighted how some proteins of the ciliary machinery contribute to temporal and spatial segregation of signalling as well as provided some questions on the importance these proteins might have in immunity.

Minor:

  • Some sentences are hard to follow in the context (for example, “At bird’s view”, supposedly “bird’s-eye view”, or the expression “speed of sound”) and need rewriting in a more scientific (less journalistic) context.

We have replace this sentence by “in brief”.

  • It is hard to understand why some words or sentences are underlined or bold.

To avoid confusion, we have eliminated all underlined and bold highlights.

  • In line 279, immunological synapse should be already written as the abbreviation IS, which was described in line 202. The manuscript should be carefully edited.

This has been corrected.

Reviewer 2 Report

This review discusses mechanisms that cells use in order to regulate signal transduction, with a focus on T cells, but also including some novel comparisons to mechanisms employed by cilia. It is very well written and engaging.

There are some issues that should be addressed prior to publication:

Major points:

  • The figure legends for figures 2 and 3 are mixed up.
  • Phosphatases such as CD45 have not been considered in this article although they play a large role in maintaining basal TCR signaling levels and dampening signaling after activation. Research shows that CD45 is not found in peripheral microclusters of TCRs, but can be found in the cSMAC (Varma R, Campi G, Yokosuka T et al. T-cell receptor-proximal signals are sustained in peripheral microclusters and terminated in the central supramolecular activation cluster. Immunity 2006; 25:117-127.) Also, negative co-stimulators like CTLA-4 could be mentioned in this context. The authors might consider consulting the 2008 Springer book: Multichain Immune Recognition Receptor Signaling: From Spatiotemporal Organization to Human Disease, edited by Alexander B. Sigalov; while several of the models proposed in this book remain unproven, significant gains have been made over the past decade that support the Permissive Geometry Model of T Cell Signaling. The recent review of Schamel, Alarcon and Minguet, Immunological Reviews, 2019; 291:8-25 should be considered and referenced.
  • Also, the maturation status of T cells and how this may influence receptor composition in the membrane, and subsequent signaling and outcomes should be addressed. For example, maturation could be included in the list given in lines 250-252 regarding structure of the immune synapse depending on T cell and APC types.
  • Signal strength plays a significant role during T cell development and should also be discussed or at least mentioned.

Minor points:

  • In Figure 1, there is no actin interactor depicted on the filopodia – is this intentional, or was this left out due to space considerations? Also, please add ions to your legend. Finally, does the chemokine receptor need to be designated as such, or would simply membrane receptor suffice, since there are many different receptors in the membrane that function similarly in terms of endo/exocytosis? This is not specified in the text or figure legend.
  • In Figures 2 and 3, please avoid using the peach colour, or make it darker. It is difficult to see. The bottom part of Figure 2 is difficult to understand, and parts a and b are not referred to in the text. As such, I would recommend removing parts a and b, and making parts c and d larger and easier to understand. The parts of the figure should be referred to in order in the text. Here fig. 2d is indicated in line 256, the c-d in line 257. Also, what is the blue part depicting? I assume this is meant to depict the nucleus, but it should be labeled or mentioned in the figure legend. The figure legend that should be with figure 2 states that there is drastic spreading of the T cell, but it does not look particularly drastic in the figure – it would be good to modify the figure as such.
  • In some places acronyms are in bold type and some words are underlined. This sometimes appears to be random, so it would be helpful for the reader if the authors explain their use of underlining and boldface at the beginning of the review. If these were unintentional, please remove them.
  • In several places the acronym is written first and the description is in brackets thereafter – this should be done in the reverse order: i.e. write out the term in full and put the acronym in brackets the first time, then use the acronym thereafter. Examples of this are in lines 132-133, 134-135 and 209.
  • While the review is very well written, replacement with more appropriate wording is suggested:
    • Line 43: replace ‘length’ with ‘duration’
    • 53: replace ‘at bird’s view’ with ‘from a bird’s eye point of view’
    • 62: replace ‘chemokine’ with ‘cytokine’ (IL-2 is a cytokine)
    • 71: replace ‘fast’ with ‘rapidly’
    • 100: replace ‘too fast’ with ‘so quickly’
    • 119: replace ‘us’ with ‘as’
    • 157: place ‘will’ before ‘the T cell’
    • 161: replace ‘length’ with ‘duration’
    • 190: replace ‘in’ with ‘over’ before ‘time’
    • 203: shift ‘anew’ to after ‘ITAMS’
    • 223: replace ‘As for’ with ‘Similar to’
    • 229: replace ‘of’ with ‘from’
    • 236: replace ‘supports’ with ‘suggests’
    • 240: replace ‘and’ with ‘, to’ after ‘resolution’
    • 261: replace ‘dispose’ with ‘deposit’
    • 325: replace ‘these’ with ‘this’
    • 332: replace ‘disposal’ with ‘dispersal’
    • 353: replace ‘f.i.’ with ‘for example’
    • 375: replace ‘their’ with ‘the’ and insert ‘of’ between ‘role’ and ‘membrane’
    • 425: remove ‘an’ and reverse the order of ‘active’ and ‘signalling’
  • Some typos that should be fixed are:
    • 23: remove comma after ‘platform’
    • 54: remove ‘s’ from ‘cells’
    • 56: add ‘s’ to ‘increase’
    • 72: add ‘s’ to ‘allow’
    • 127: add ‘d’ to ‘localise’
    • 135: add ‘ing’ to stretch
    • 136: remove comma
    • 137: remove ‘s’ from ‘cells’
    • 162: add ‘such’ before ‘as’
    • 184: add ‘to’ before ‘TCR’, and ‘%’ after ‘2’
    • 185: remove ‘between reports’
    • 191: remove ‘a’ after ‘require’
    • 199: insert ‘the’ after ‘of’
    • 219: change ‘analogue’ to ‘analogous’
    • 238: insert ‘a’ between ‘as’ and ‘signalling’
    • 283: insert ‘it’ before ‘is’ and add an ‘s’ to ‘relocate’
    • 291: add an ‘s’ to ‘form’
    • 348: remove ‘are’

Author Response

Reviewer 2:

This review discusses mechanisms that cells use in order to regulate signal transduction, with a focus on T cells, but also including some novel comparisons to mechanisms employed by cilia. It is very well written and engaging.

There are some issues that should be addressed prior to publication:

Major points:

  • The figure legends for figures 2 and 3 are mixed up.

This has now been fixed. Also, we have reorganised the figures for the sake of simplicity and combined the original figures into the new figure 1 and figure 2.

  • Phosphatases such as CD45 have not been considered in this article although they play a large role in maintaining basal TCR signaling levels and dampening signaling after activation. Research shows that CD45 is not found in peripheral microclusters of TCRs, but can be found in the cSMAC (Varma R, Campi G, Yokosuka T et al. T-cell receptor-proximal signals are sustained in peripheral microclusters and terminated in the central supramolecular activation cluster. Immunity 2006; 25:117-127.) Also, negative co-stimulators like CTLA-4 could be mentioned in this context. The authors might consider consulting the 2008 Springer book: Multichain Immune Recognition Receptor Signaling: From Spatiotemporal Organization to Human Disease, edited by Alexander B. Sigalov; while several of the models proposed in this book remain unproven, significant gains have been made over the past decade that support the Permissive Geometry Model of T Cell Signaling. The recent review of Schamel, Alarcon and Minguet, Immunological Reviews, 2019; 291:8-25 should be considered and referenced.

We appreciate the suggestions of the reviewer to improve the manuscript. We now cover these new topics in the manuscript:

  • We have included CD45 in the sequence towards T cell activation (lines 52) and in the spatial segregation section 4 (lines 337-339).
  • CTLA-4 is now referenced in section 3 (lines 207-208) in the context of co-stimulatory receptors.
  • We have added the reference suggested: Schamel et al 2019 to section 2 (line 157) providing additional reviews for the reader to consider and acquire a wider context of TCR activation models.
  • Also, the maturation status of T cells and how this may influence receptor composition in the membrane, and subsequent signaling and outcomes should be addressed. For example, maturation could be included in the list given in lines 250-252 regarding structure of the immune synapse depending on T cell and APC types.

We thank the reviewer for the constructive comment. We have added T cell maturation to the list in lines 250-252 (now located in lines 313-315, section 3) and provided as an example the differential localisation of CD3zin thymocytes vs mature T cells.

  • Signal strength plays a significant role during T cell development and should also be discussed or at least mentioned.

We appreciate the suggestion of the reviewer and have added a brief comment in section 2 (lines 185-188) addressing the correlation of signal strength and lifetime of T cell-APC interaction and cell differentiation/fate.

Minor points:

  • In Figure 1, there is no actin interactor depicted on the filopodia – is this intentional, or was this left out due to space considerations? Also, please add ions to your legend. Finally, does the chemokine receptor need to be designated as such, or would simply membrane receptor suffice, since there are many different receptors in the membrane that function similarly in terms of endo/exocytosis? This is not specified in the text or figure legend.

This figured has been removed and some of its contents have been transferred to figure 2. The role of the actin cytoskeleton has been thoroughly reviewed before so we have removed unnecessary information not needed to support the text. We have considered these comments in new figure 1 and its legend.

  • In Figures 2 and 3, please avoid using the peach colour, or make it darker. It is difficult to see. The bottom part of Figure 2 is difficult to understand, and parts a and b are not referred to in the text. As such, I would recommend removing parts a and b, and making parts c and d larger and easier to understand. The parts of the figure should be referred to in order in the text. Here fig. 2d is indicated in line 256, the c-d in line 257. Also, what is the blue part depicting? I assume this is meant to depict the nucleus, but it should be labeled or mentioned in the figure legend. The figure legend that should be with figure 2 states that there is drastic spreading of the T cell, but it does not look particularly drastic in the figure – it would be good to modify the figure as such.

The colour has been modified as suggested by the reviewer. Now the nucleus is clearly identified. To simplify and clarify figures 2 and 3 we have merged them into new figure 2, removed any unnecessary information and focused on the old figure 2c-d.

  • In some places acronyms are in bold type and some words are underlined. This sometimes appears to be random, so it would be helpful for the reader if the authors explain their use of underlining and boldface at the beginning of the review. If these were unintentional, please remove them.

Underlying and bold highlighting has been removed to avoid confusion.

  • In several places the acronym is written first and the description is in brackets thereafter – this should be done in the reverse order: i.e. write out the term in full and put the acronym in brackets the first time, then use the acronym thereafter. Examples of this are in lines 132-133, 134-135 and 209.

This has now been corrected and all acronyms are accompanied by a description in the order IJMS requires (full name + (acronym)).

  • While the review is very well written, replacement with more appropriate wording is suggested:

All suggestions have been corrected except “127: add ‘d’ to ‘localise’” since we intended to use the present form in this case.

  • Line 43: replace ‘length’ with ‘duration’
  • 53: replace ‘at bird’s view’ with ‘from a bird’s eye point of view’
  • 62: replace ‘chemokine’ with ‘cytokine’ (IL-2 is a cytokine)
  • 71: replace ‘fast’ with ‘rapidly’
  • 100: replace ‘too fast’ with ‘so quickly’
  • 119: replace ‘us’ with ‘as’
  • 157: place ‘will’ before ‘the T cell’
  • 161: replace ‘length’ with ‘duration’
  • 190: replace ‘in’ with ‘over’ before ‘time’
  • 203: shift ‘anew’ to after ‘ITAMS’
  • 223: replace ‘As for’ with ‘Similar to’
  • 229: replace ‘of’ with ‘from’
  • 236: replace ‘supports’ with ‘suggests’
  • 240: replace ‘and’ with ‘, to’ after ‘resolution’
  • 261: replace ‘dispose’ with ‘deposit’
  • 325: replace ‘these’ with ‘this’
  • 332: replace ‘disposal’ with ‘dispersal’
  • 353: replace ‘f.i.’ with ‘for example’
  • 375: replace ‘their’ with ‘the’ and insert ‘of’ between ‘role’ and ‘membrane’
  • 425: remove ‘an’ and reverse the order of ‘active’ and ‘signalling’

Some typos that should be fixed are:

  • 23: remove comma after ‘platform’
  • 54: remove ‘s’ from ‘cells’
  • 56: add ‘s’ to ‘increase’
  • 72: add ‘s’ to ‘allow’
  • 127: add ‘d’ to ‘localise’ (I intended to use the present form)
  • 135: add ‘ing’ to stretch
  • 136: remove comma
  • 137: remove ‘s’ from ‘cells’
  • 162: add ‘such’ before ‘as’
  • 184: add ‘to’ before ‘TCR’, and ‘%’ after ‘2’
  • 185: remove ‘between reports’
  • 191: remove ‘a’ after ‘require’
  • 199: insert ‘the’ after ‘of’
  • 219: change ‘analogue’ to ‘analogous’
  • 238: insert ‘a’ between ‘as’ and ‘signalling’
  • 283: insert ‘it’ before ‘is’ and add an ‘s’ to ‘relocate’
  • 291: add an ‘s’ to ‘form’
  • 348: remove ‘are’

Reviewer 3 Report

After a very good introduction, the review becomes a bit confusing. The original hypothesis (the “where and when” is central to the triggering, regulation and outcome of T cell activation) is not really discussed any more. More worryingly, there are many inaccuracies regarding T cell biology and many references are missing or unsuitably used.

More specifically:

Line 59: It’s not recognition of TCR and pMHC, this is inaccurate. It is the recognition of a cognate peptide, presented by MHC, by TCR. This has to be corrected.

Line 60: whether Lck is activated or not upon engagement of TCR is a very debated topic (see Nika et al, 2010, Stirnweiss et al, 2013). Unless the authors want to discuss this point extensively, they should just mention something like “TCR engagement leads to the phosphorylation of the TCR-CD3 complex intracellular domains by Lck”

Line 60: Lck and Fyn do not only phosphorylate zeta, but also the other CD3 subunits (gamma, delta, epsilon). Again, it might be easier to say to say something like “phosphorylate the intracellular domains of the TCR-CD3 complex”.

Also, Lck does phosphorylate Zap70, but only after this one has been recruited to pCD3 (in fact the primary outcome of pCD3 is to recruit Zap70), this should be obvious in the text and has to be corrected.

Line 62: the sentence is unclear and not elegant: …that…that…

Line 61: it is inaccurate to mention LAT as being activated, as there is no conformational change or enzymatic activity. A more suitable word should be chosen. Lat possesses multiple phosphorylation sites, that acts as docking site for effector (as rightfully mentioned later).

Lines 52-66: globally, the early steps of T cell activation are rather clumsily summarized in this paragraph. It would be good to cite at least one review of a specialist in the field for the readers who want to have a more comprehensive picture.

Line 71: the “fast” is a bit redundant with the previous sentence (speed of sound, quickly, miliseconds…).

Line 72: the notion that there is a “final target” to signalling is disputable. I don’t think we have a system where 1 signal is paired with 1 target. This should be corrected (or discussed).

Line 76: how exactly does pH affects mechanotransduction? It is not mentioned at all in the ref. 9 used here. It’s also a bit disturbing to notice that the ref. is about membrane thickness, which is about the only physical property not mentioned in line 76-77… This has to be corrected (and proper refs. used).

Line 99: which activation process are we talking about?

Line 100: what do the author means by very early steps and full-fledged activation? In terms of time scale? And “too fast” for what? Do the author imply it is happening so fast that we cannot see it!? There are many approaches that allow ms time scale measurements. This sentence is highly inaccurate and needs rewriting.

Lines 102-107: I am not so sure that all these molecules can be found at the tip of microvilli. The authors failed to support their claim with any reference. As for presenting TCR through the glycocalyx, it’s an interesting hypothesis, but, again, the authors present no evidence for it (again, the references 14 and 15 are reviews, and they do not really support the author’s claim). The proper references to discuss this theme are

  • Ghosh et al, 2020
  • Razvag et al, 2018
  • Cai et al., 2017
  • Jung et al, 2016

Line 107: How do integrins promote TCR-pMCH encounter exactly? I am not aware of such mechanism, and again, there is no proper citations.

Line 111-12: This sentence is inaccurate (the dissociation rate is not increased). The findings of Kong et al is that force prolonged bond lifetimes of integrins, in a process called “catch bond”.

Line 114: Again, it seems the authors have been a bit quick when reading the referenced paper: it’s not about responding “faster”, but how the lifetime of the bound is prolonged (and how these properties persist when there is no force any more). This has to be corrected.

Line 144-145: what do the authors mean by “cooperate with Ag-mediated TCR activation”? And again, references are missing here, especially when numbers are given (two orders of magnitude)

Line 151: it’ not quite clear what are the “aforementioned mechanisms”… Should be specified.

Line 157: Ref is missing for TCR tickling…

Line 157: This is again inaccurate, as this is totally ignoring co-receptors. For instance, T cell activated by strong agonist in absence of CD28-mediated signalling will become anergic…

Line 67-159:!!! How this paragraph on mechanotransduction is relevant to the topic of the review (spatiotemp segregation of signalling) is really unclear, and certainly explained nowhere! How local can be mechanical stimuli, and how they can act locally in T cells, to regulate or promote specific signal events related to T cell activation is not described properly (if at all).

Line 175: Lck is the kinase that starts the whole TCR signalling cascade. Without Lck, no T cell activation. The reference cited only shows that Fyn can compensate for Lck if it is KO. Not that Lck can reduce the activation threshold. This sentence makes no sense and has to be removed.

Line 199: actually, the intracellular trafficking of Lck is quite different from that of TCR. Here again, no ref to back this claim up.

Line 204-205: how is “peripheral Lck” supposed to reinforce TCR signalling, when Lck is required for TCR signalling? How does Lck promote a mature (?) conformation of integrins? Here again, the two reference (40-41) are totally out of place and do not support the claim a single bit.

I unfortunately cannot keep going that way, this is far too much work. This review needs extensive revision. It needs a structure, a message (that would not only be within the title), and most importantly, it needs proper referencing.

And, the figures are also quite unclear and do not help the understanding of the review at all. What is the purpose of figure 1, what is the reader supposed to gather from it? Typically, figure 1, what is this mechanical pressure, distributed all over the cell, and having the same direction and intensity on every point? What are the consequences of the forces on the proteins represented? How is force acting on GPCRs? And chemokine Rs? (btw chemokine Rs are GPCRs!). How is actin transferring force? How is the lamellipodia involved in mediating force? What are these mysterious “actin interactors?” Why is there an ion channel at the tip of the microvillus? And TCR is not even on the figure! And so, on the right, chemical signalling relies only on endocytic trafficking? How is intracellular trafficking supposed to be connected to regulation of gene expression?

Same for fig. 3. The legend is not even matching the figure. What is the reader supposed to learn from this figure? What's the message, concept, mechanism that is depicted here? 

Author Response

After a very good introduction, the review becomes a bit confusing. The original hypothesis (the “where and when” is central to the triggering, regulation and outcome of T cell activation) is not really discussed any more. More worryingly, there are many inaccuracies regarding T cell biology and many references are missing or unsuitably used.

More specifically:

Line 59: It’s not recognition of TCR and pMHC, this is inaccurate. It is the recognition of a cognate peptide, presented by MHC, by TCR. This has to be corrected.

We appreciate the reviewer pointing this out. We have been amended this to avoid confusion (now in lines 47-49).

Line 60: whether Lck is activated or not upon engagement of TCR is a very debated topic (see Nika et al, 2010, Stirnweiss et al, 2013). Unless the authors want to discuss this point extensively, they should just mention something like “TCR engagement leads to the phosphorylation of the TCR-CD3 complex intracellular domains by Lck”

We understand this can lead to confusion so we have corrected it (now in lines 49-52).

Line 60: Lck and Fyn do not only phosphorylate zeta, but also the other CD3 subunits (gamma, delta, epsilon). Again, it might be easier to say to say something like “phosphorylate the intracellular domains of the TCR-CD3 complex”.

We have added the reviewer’s suggestion (now in lines 50-51).

Also, Lck does phosphorylate Zap70, but only after this one has been recruited to pCD3 (in fact the primary outcome of pCD3 is to recruit Zap70), this should be obvious in the text and has to be corrected.

This has been corrected (lines 53-55)

Line 62: the sentence is unclear and not elegant: …that…that…

This has now been re-phrased to avoid repetition.

Line 61: it is inaccurate to mention LAT as being activated, as there is no conformational change or enzymatic activity. A more suitable word should be chosen. Lat possesses multiple phosphorylation sites, that acts as docking site for effector (as rightfully mentioned later).

We understand this can be confusing and it has been amended (now in lines 54-58).

Lines 52-66: globally, the early steps of T cell activation are rather clumsily summarized in this paragraph. It would be good to cite at least one review of a specialist in the field for the readers who want to have a more comprehensive picture.

We appreciate the suggestion of the reviewer, to present the signalling steps in a clearer way we have edited this part, it has been moved to the introduction and added some information. We have also added some references of reviews to be taken account by the reader to have a bigger picture. (now in lines 39-64).

Line 71: the “fast” is a bit redundant with the previous sentence (speed of sound, quickly, miliseconds…).

This has been reorganised to avoid redundancy (now in lines 102-103).

Line 72: the notion that there is a “final target” to signalling is disputable. I don’t think we have a system where 1 signal is paired with 1 target. This should be corrected (or discussed).

We have removed “final” to avoid inaccuracies (now in line 104).

Line 76: how exactly does pH affects mechanotransduction? It is not mentioned at all in the ref. 9 used here. It’s also a bit disturbing to notice that the ref. is about membrane thickness, which is about the only physical property not mentioned in line 76-77… This has to be corrected (and proper refs. used).

We thank the reviewer for pointing this out. We originally referred to how pH and electrostatic charges affect membrane voltage and therefore activity of voltage/ion-channels. We have included adequate references that address each of the listed factors and clarified the ideas in the text to avoid confusion (now in lines 108-110).

Line 99: which activation process are we talking about?

We have now specified which process (now in line 117).

Line 100: what do the author means by very early steps and full-fledged activation? In terms of time scale? And “too fast” for what? Do the author imply it is happening so fast that we cannot see it!? There are many approaches that allow ms time scale measurements. This sentence is highly inaccurate and needs rewriting.

We have re-phrased this sentence to be more specific (now in lines 117-119).

Lines 102-107: I am not so sure that all these molecules can be found at the tip of microvilli. The authors failed to support their claim with any reference. As for presenting TCR through the glycocalyx, it’s an interesting hypothesis, but, again, the authors present no evidence for it (again, the references 14 and 15 are reviews, and they do not really support the author’s claim). The proper references to discuss this theme are:

  • Ghosh et al, 2020
  • Razvag et al, 2018
  • Cai et al., 2017
  • Jung et al, 2016

We have added the suggested references and re-formulated our statement regarding the accessibility to TCR from a mere hypothesis to a more robust (and yet only potential) mechanism (now in lines 125-129).

Line 107: How do integrins promote TCR-pMCH encounter exactly? I am not aware of such mechanism, and again, there is no proper citations.

We appreciate the reviewer for pointing out that this sentence is unclear, we have amended the text and added the appropriate references (now in lines 125-129).

Line 111-12: This sentence is inaccurate (the dissociation rate is not increased). The findings of Kong et al is that force prolonged bond lifetimes of integrins, in a process called “catch bond”.

We thank the reviewer for pointing this out that could be misinterpreted. Instead of refer to the dissociation rate (Kd) we have substituted it for “lifetime bond” to make it more accessible for the reader (now in line 135).

Line 114: Again, it seems the authors have been a bit quick when reading the referenced paper: it’s not about responding “faster”, but how the lifetime of the bound is prolonged (and how these properties persist when there is no force any more). This has to be corrected.

We appreciate this comment and agree with the reviewer that the article referenced states indeed that the lifetime of the bound is prolonged. The article also described the memory capabilities of primed adhesions that would achieve to longer lifetime bonds after repetitive stimulation and even after cease of the mechanical stimulus. Since this was not clearly described in the text we have clarified this to avoid confusion (now in lines 137-139).

Line 144-145: what do the authors mean by “cooperate with Ag-mediated TCR activation”? And again, references are missing here, especially when numbers are given (two orders of magnitude)

We understand this can cause misunderstanding and therefore we have modified the text and placed the referenced appropriately (lines 161-165).

Line 151: it’ not quite clear what are the “aforementioned mechanisms”… Should be specified.

We have now specified these mechanisms (lines 171-173).

Line 157: Ref is missing for TCR tickling…

We have added the reference (lines 178-179).

Line 157: This is again inaccurate, as this is totally ignoring co-receptors. For instance, T cell activated by strong agonist in absence of CD28-mediated signalling will become anergic…

We have removed this sentence to avoid inaccuracies (lines 179-180).

Line 67-159:!!! How this paragraph on mechanotransduction is relevant to the topic of the review (spatiotemp segregation of signalling) is really unclear, and certainly explained nowhere! How local can be mechanical stimuli, and how they can act locally in T cells, to regulate or promote specific signal events related to T cell activation is not described properly (if at all).

We appreciate the suggestions of the reviewer and have rearranged this section aiming to clarify the relevance of mechanical cues in the temporal segregation of signals. We have better explain in the manuscript that mechanical signals are particularly relevant during the very first steps of T cell-APC interaction, where very fast rearrangement of cell and molecules are required to stablish a more stable interaction later on. We have included further references to support this idea.

Line 175: Lck is the kinase that starts the whole TCR signalling cascade. Without Lck, no T cell activation. The reference cited only shows that Fyn can compensate for Lck if it is KO. Not that Lck can reduce the activation threshold. This sentence makes no sense and has to be removed.

We understand the concern of the reviewer since the way we originally wrote this could lead to confusion. We have clarified in the manuscript that these observations were made in LCK KO reconstituted with FYN. Still, we consider that, explained properly, the article we reference here is relevant to debate the potential role of LCK (but not FYN) in modifying the threshold of activation in T cells (Lines 211).

Line 199: actually, the intracellular trafficking of Lck is quite different from that of TCR. Here again, no ref to back this claim up.

We were referring to both molecules sharing some recycling pathways like Rab11 but we agree with the reviewer that this is inaccurate and have removed it (lines 237-238).

Line 204-205: how is “peripheral Lck” supposed to reinforce TCR signalling, when Lck is required for TCR signalling? How does Lck promote a mature (?) conformation of integrins? Here again, the two reference (40-41) are totally out of place and do not support the claim a single bit.

We have removed this sentence (244-246).

I unfortunately cannot keep going that way, this is far too much work. This review needs extensive revision. It needs a structure, a message (that would not only be within the title), and most importantly, it needs proper referencing.

We appreciate the time spent by the reviewer as well as all the constructive suggestions to improve the manuscript. We have extensively reviewed the manuscript to improve the structure, clarity and order of ideas. More specifically:

  • We have expanded the introduction to better introduce the molecules that will be discussed later on.
  • We have removed non-essential or repetitive ideas to simplify the manuscript and making it more compelling.
  • We have stressed the main ideas of the manuscript to help the reader follow the story.
  • We have organised the sections more clearly and added subsections 3.1 and 3.2 to better describe the section of spatial segregation of signalling.
  • We have revised the references and added new references to improve the focus of the manuscript.

And, the figures are also quite unclear and do not help the understanding of the review at all. What is the purpose of figure 1, what is the reader supposed to gather from it? Typically, figure 1, what is this mechanical pressure, distributed all over the cell, and having the same direction and intensity on every point? What are the consequences of the forces on the proteins represented? How is force acting on GPCRs? And chemokine Rs? (btw chemokine Rs are GPCRs!). How is actin transferring force? How is the lamellipodia involved in mediating force? What are these mysterious “actin interactors?” Why is there an ion channel at the tip of the microvillus? And TCR is not even on the figure! And so, on the right, chemical signalling relies only on endocytic trafficking? How is intracellular trafficking supposed to be connected to regulation of gene expression?

We agree with the reviewer that the original figure poorly supported the main text. We have simplified the contents of the figures, now we show 2 figures instead of 3 and have focused only on T cell signalling. The new distribution is more concise and better supports the text, following and clear order in parallel to the main text.

Same for fig. 3. The legend is not even matching the figure. What is the reader supposed to learn from this figure? What's the message, concept, mechanism that is depicted here?

We have corrected this mistake and updated the text of figure legends to fit the new figures.

Round 2

Reviewer 3 Report

The authors did a great work to be more consistant in the way they support claims by citations. And cleaned up the text a bit. 

I just have a few more comments, as I have been reading further down the text. 

Line 218: I presume the authors mean phosphorylation of Y394?

Line 234: ref. 72 is a review. It would be more suitable to cite the actual research articles (or at least some of them).

Line 247-254 is one very long sentence, that is very hard to read. And Lck is not internalised by Rab11+ vesicles (because it’s a recycling Rab, nothing is internalised in Rab11+ vesicles), this has to be corrected.

Line 254: The mechanism described here is not very clear. Does UNC119 extract Lck from PM or endosomes into some UNC119+ endosome? If yes, of which nature are these endosomes (endocytic, sorting, recycling?). Or is Lck extracted in the cytosol (that would be unexpected, because there is not much Lck in the cytosol in T cells)? And why would UNC119 extract Lck from Rab11 endosomes, which are anyway bringing Lck to the IS?

Line 263: which diffusion barriers are the authors referring to? Are they studies demonstrating le presence of diffusion barrier in the IS, like it’s done for the macrocytic cup for instance? If yes, these references should be added (if not, the diffusion barriers should be removed)

Line 271-2: Ref for CME of Lat is missing.

Line 273: Ref for delivery of Lat to PM is missing     

Line 279: It’s not clear how the centrosome would coordinate MT-based transport. The centrosome is at the base of the MT network, while transport of intracellular compartments/vesicles is achieved by MT motors. How is the activity of these motors coordinated by the centrosome?

Line 283: The authors should define what they mean by exhausted and non-exhausted signalling molecules (is it de-P, or in a GDP/ADP state?)

Line 285: Ref 80 does not really support that claim (whereas indeed 81 an 82 do).

Line 287: The message is somehow confusing, as intracellular trafficking needs a bit of time to kick in in T cells. Besides, it seems intuitive that the very early steps of T cell activation rely on signalling molecules that are on the spot when TCR is triggered, isn’t it? How a molecule that is in a vesicle “en route” to the IS, or that needs to be packaged into an endocytic vesicle (which takes 1-3 min) can contribute to what happens milliseconds or seconds after TCR is triggered? Accordingly, people believing in the role of intracellular trafficking in T cells rather postulate that it would contribute to late/sustained signalling. This conclusion should really be re-thought (and rewritten, I get what the authors tried to mean, but “centrist” means someone at the centre of the political spectrum… And “focused on earlier timepoints and to high temporal resolution” is not proper English).

Line 306: It’s not clear what the authors means by “Membrane domains are more dynamic than cellular organelles”. Organelle, especially of the intracellular trafficking network, can be extremely dynamics, undergoing fission/fusion events almost continuously. While membrane domains can be quite stable, anchored to cortical actin, or reinforced by protein clustering. As there is no reference to support this puzzling claim, it would be better to remove it.

Line 320: The authors did great in mentioning that the bull eye pattern is the most studied one, but they should mention the fact that T cell IS with dendritic cells are totally different multiclustered.

Line 321: formation of the actin ring relies mostly on Arp2/3 mediated actin nucleation, and maybe later on MIIA. This should be mentioned here (could cite an excellent review: https://www.annualreviews.org/doi/10.1146/annurev-immunol-042718-041341 )

Line 358: This sentence is highly unclear. DAG relocates the MTOC independently of TCR, do you mean TCR complex does not contribute physically to MTOC relocation? That would be very surprising anyway! Because if MTOC relocation is downstream of TCR triggering, then, it cannot happen without TCR being at least a little bit involved upstream, can'it?

Line 398: If disposal was not the right word (!), dispersal is not either. Distribution might be what you are looking for (in case of any doubt about the meaning of a word, I strongly recommend using Google.)

Author Response

The authors did a great work to be more consistant in the way they support claims by citations. And cleaned up the text a bit. 

I just have a few more comments, as I have been reading further down the text. 

Line 218: I presume the authors mean phosphorylation of Y394?

We thank the review for pointing this out, we have now specified the phosphorylated residue of LCK.

Line 234: ref. 72 is a review. It would be more suitable to cite the actual research articles (or at least some of them).

We have now replaced reference 72 for references 72-75.

Line 247-254 is one very long sentence, that is very hard to read. And Lck is not internalised by Rab11+ vesicles (because it’s a recycling Rab, nothing is internalised in Rab11+ vesicles), this has to be corrected.

We appreciate the comment of the reviewer. This sentence has been edited so it is easier to read and understand, and we have corrected the information regarding Rab11.

Line 254: The mechanism described here is not very clear. Does UNC119 extract Lck from PM or endosomes into some UNC119+ endosome? If yes, of which nature are these endosomes (endocytic, sorting, recycling?). Or is Lck extracted in the cytosol (that would be unexpected, because there is not much Lck in the cytosol in T cells)? And why would UNC119 extract Lck from Rab11 endosomes, which are anyway bringing Lck to the IS?

We recognise that the mechanism described is not clear. There are no evidences regarding which vesicles could be involved in this mechanism, but we still propose the hypothesis that UNC119 could extract LCK from other areas of the plasma membrane, from endomembranes or from both. In the case of UNC119 extracting LCK from inner vesicles, we discuss in the manuscript that this could help keeping the homeostasis of the plasma membrane by reducing the rate of membrane fusion involved in vesicle-mediated trafficking of LCK. We have modified the text aiming to express this more clearly.

Line 263: which diffusion barriers are the authors referring to? Are they studies demonstrating le presence of diffusion barrier in the IS, like it’s done for the macrocytic cup for instance? If yes, these references should be added (if not, the diffusion barriers should be removed)

We agree that due to editions of the manuscript this is not relevant in this part of the manuscript anymore and we have moved it to section three where diffusion barriers are thoroughly discussed (lines 429-432).

Line 271-2: Ref for CME of Lat is missing.

Line 273: Ref for delivery of Lat to PM is missing     

We thank the reviewer for pointing out the missing references and have updated the manuscript with the pertinent references.

Line 279: It’s not clear how the centrosome would coordinate MT-based transport. The centrosome is at the base of the MT network, while transport of intracellular compartments/vesicles is achieved by MT motors. How is the activity of these motors coordinated by the centrosome?

We agree that the reference to the centrosome is poorly explained. To properly address this we would need to expand the discussion considerably so we have decided to remove the reference to the centrosome since it does not provide key information to the topic discussed.

Line 283: The authors should define what they mean by exhausted and non-exhausted signalling molecules (is it de-P, or in a GDP/ADP state?)

We understand the confusion this sentence can lead to so we have defined exhaustion in line 264.

Line 285: Ref 80 does not really support that claim (whereas indeed 81 an 82 do).

We have removed reference 80.

Line 287: The message is somehow confusing, as intracellular trafficking needs a bit of time to kick in in T cells. Besides, it seems intuitive that the very early steps of T cell activation rely on signalling molecules that are on the spot when TCR is triggered, isn’t it? How a molecule that is in a vesicle “en route” to the IS, or that needs to be packaged into an endocytic vesicle (which takes 1-3 min) can contribute to what happens milliseconds or seconds after TCR is triggered? Accordingly, people believing in the role of intracellular trafficking in T cells rather postulate that it would contribute to late/sustained signalling. This conclusion should really be re-thought (and rewritten, I get what the authors tried to mean, but “centrist” means someone at the centre of the political spectrum… And “focused on earlier timepoints and to high temporal resolution” is not proper English).

We appreciate the suggestion of the reviewer and we have amended the text with more appropriate wording to avoid confusion.

Line 306: It’s not clear what the authors means by “Membrane domains are more dynamic than cellular organelles”. Organelle, especially of the intracellular trafficking network, can be extremely dynamics, undergoing fission/fusion events almost continuously. While membrane domains can be quite stable, anchored to cortical actin, or reinforced by protein clustering. As there is no reference to support this puzzling claim, it would be better to remove it.

We have now removed the sentence as requested.

Line 320: The authors did great in mentioning that the bull eye pattern is the most studied one, but they should mention the fact that T cell IS with dendritic cells are totally different multiclustered.

We appreciate the suggestion; we point the reader to a couple of reviews where they discuss other types of synapses including those of dendritic cells.

Line 321: formation of the actin ring relies mostly on Arp2/3 mediated actin nucleation, and maybe later on MIIA. This should be mentioned here (could cite an excellent review: https://www.annualreviews.org/doi/10.1146/annurev-immunol-042718-041341 )

We thank the reviewer for pointing this out. We have added the suggested review to provide the reader with further information on how actin and actomyosin contribute to the immunological synapse.

Line 358: This sentence is highly unclear. DAG relocates the MTOC independently of TCR, do you mean TCR complex does not contribute physically to MTOC relocation? That would be very surprising anyway! Because if MTOC relocation is downstream of TCR triggering, then, it cannot happen without TCR being at least a little bit involved upstream, can'it?

We thank the reviewer for pointing this out, we have amended this part to avoid confusion.

Line 398: If disposal was not the right word (!), dispersal is not either. Distribution might be what you are looking for (in case of any doubt about the meaning of a word, I strongly recommend using Google.)

We have corrected this.